# Basal VEGF-A and ACE Plasma Levels of Metastatic Colorectal Cancer Patients Have Prognostic Value for First-Line Treatment with Chemotherapy Plus Bevacizumab

**DOI:** 10.3390/cancers14133054

**Published:** 2022-06-21

**Authors:** M. José Ortiz-Morales, Marta Toledano-Fonseca, Rafael Mena-Osuna, M. Teresa Cano, Auxiliadora Gómez-España, Juan R. De la Haba-Rodríguez, Antonio Rodríguez-Ariza, Enrique Aranda

**Affiliations:** 1Medical Oncology Department, Reina Sofía University Hospital, 14004 Córdoba, Spain; mariaj.ortiz.morales.sspa@juntadeandalucia.es (M.J.O.-M.); mteresa.cano.sspa@juntadeandalucia.es (M.T.C.); auxiliadora.gomez.sspa@juntadeandalucia.es (A.G.-E.); juanr.delahaba.sspa@juntadeandalucia.es (J.R.D.l.H.-R.); earandaa@seom.org (E.A.); 2Maimonides Biomedical Research Institute of Cordoba (IMIBIC), 14004 Córdoba, Spain; marta.toledano@imibic.org (M.T.-F.); b72meosr@uco.es (R.M.-O.); 3Department of Medicine, Faculty of Medicine, University of Córdoba, 14004 Córdoba, Spain; 4Cancer Network Biomedical Research Centre (CIBERONC), 28029 Madrid, Spain

**Keywords:** VEGF-A, angiotensin-converting enzyme, colorectal cancer, anti-angiogenic drug, bevacizumab, prognosis, biomarker

## Abstract

**Simple Summary:**

Molecular biology knowledge has enabled the incorporation of targeted therapies, such as the anti-angiogenic drug bevacizumab, into combined chemotherapy regimens for the treatment of metastatic colorectal cancer. However, to date, there are no reliable useful biomarkers to predict the efficacy of this anti-angiogenic therapy. The objective of this prospective study was to evaluate potential circulating plasma biomarkers in mCRC patients prior to the start of first-line treatment with chemotherapy plus bevacizumab. We found that high VEGF-A and low ACE plasma levels were associated with poor OS after treatment. Moreover, a simple scoring system combining both biomarkers efficiently stratified patients into high- or low-risk groups, which allows the selection of patients for anti-angiogenic therapy.

**Abstract:**

The identification of factors that respond to anti-angiogenic therapy would represent a significant advance in the therapeutic management of metastatic-colorectal-cancer (mCRC) patients. We previously reported the relevance of VEGF-A and some components of the renin–angiotensin-aldosterone system (RAAS) in the response to anti-angiogenic therapy in cancer patients. Therefore, this prospective study aims to evaluate the prognostic value of basal plasma levels of VEGF-A and angiotensin-converting enzyme (ACE) in 73 mCRC patients who were to receive bevacizumab-based therapies as a first-line treatment. We found that high basal VEGF-A plasma levels were significantly associated with worse overall survival (OS) and progression-free survival (FPS). On the other hand, low ACE levels were significantly associated with poor OS. Importantly, a simple scoring system combining the basal plasma levels of VEGF-A and ACE efficiently stratified mCRC patients, according to OS, into high-risk or low-risk groups, prior to their treatment with bevacizumab. In conclusion, our study supports that VEGF-A and ACE may be potential biomarkers for selecting those mCRC patients who will most benefit from receiving chemotherapy plus bevacizumab treatment in first-line therapy. Additionally, our data reinforce the notion of a close association between the RAAS and the anti-angiogenic response in cancer.

## 1. Introduction

Colorectal cancer is the second leading cause of cancer deaths in men and women worldwide, with an estimated incidence of 1.9 million new cases diagnosed in 2020 [1]. Survival rates for this disease depend on clinical, biological, and molecular prognostic factors, with median overall survival (OS) in metastatic disease exceeding 30 months [2,3].

The treatment of metastatic colorectal cancer (mCRC) requires multidisciplinary management, and molecular biology knowledge has enabled the incorporation of targeted therapies, such as anti-EGFR and anti-angiogenic drugs, into combined chemotherapy regimens [4,5]. In this regard, bevacizumab is a recombinant humanized monoclonal antibody that inhibits vascular endothelial growth factor (VEGF-A), which is important for angiogenesis signaling, commonly upregulated in mCRC [6,7]. Tumor progression in mCRC involves multiple molecular factors that modify the processes of cell proliferation, differentiation, and death [8]. In this context, the renin–angiotensin system (RAAS), which plays an important role in the relationships between the tumor microenvironment, the vasculature, and the immune system, has also been reported to participate in the process of tumor angiogenesis [9].

In the era of personalized medicine, the role of KRAS/NRAS mutational status as a predictor of resistance to anti-EGFR treatment in mCRC is widely known [10,11]. Likewise, in recent years, new predictive biomarkers of response in mCRC have emerged, such as microsatellite instability–high (MSI-H)/mismatch repair deficiency (dMMR) in immunotherapy [12]; mutation in BRAFV600E in the combination of anti-BRAF and anti-EGFR treatments [13]; and HER2 overexpression in anti-HER2 therapy [14]. However, to date, there are no reliable useful biomarkers to predict the efficacy of anti-angiogenic therapy in mCRC. Numerous studies have measured angiogenic factors in plasma and/or tumor tissue, to select the subgroup of patients who might benefit most [15]. Although there is evidence of a prognostic role for VEGF-A, no consistent data have been reported as a predictive factor, explaining why this biomarker has not been implemented in clinical practice [16,17,18,19,20].

Shedding light on these discrepancies, our group recently reported the results of the POLAF clinical trial [21], which supports the efficacy of FOLFIRI plus the anti-angiogenic drug aflibercept as a second-line treatment in mCRC, after the failure of oxaliplatin-based therapy; this suggests VEGF-A as a potential biomarker to predict better outcomes. On the other hand, we also previously reported that higher circulating levels of the angiotensin-converting enzyme (ACE) is associated with a better response to anti-angiogenic treatment with bevacizumab in breast- and colorectal-cancer patients [22]. 

Therefore, the objective of this study was to evaluate the prognostic value of basal VEGFA and ACE plasma levels of mCRC patients prior to the start of first-line treatment with chemotherapy plus bevacizumab.

## 2. Materials and Methods

### 2.1. Patients

This is a prospective longitudinal observational study, with a total of 200 patients consecutively assessed before the initiation of standard first-line bevacizumab-based treatment, from March 2017 to December 2020, at the medical oncology department of the Reina Sofia University Hospital, Cordoba (Spain).

The inclusion criteria were: age > 18 years; histological confirmation of unresectable mCRC; ECOG < 2; good biochemical and hematological function; indication for first-line treatment including chemotherapy with bevacizumab. The exclusion criteria were: previous synchronous or metachronous neoplasms; ECOG ≥ 2; resectable disease; impossibility to evaluate response to treatment; indication for chemotherapy alone and/or added to anti-EGFR; no indication for bevacizumab due to uncontrolled arterial hypertension and/or proteinuria and/or high risk of bleeding; and impossibility of determining VEGF-A or ACE plasma levels.

Peripheral blood samples were collected from each patient prior to the administration of chemotherapy plus bevacizumab. Patients should have disease measurable by RECIST criteria [23], and response assessments were performed according to clinical practice, using CT scan, every three months and/or six cycles from the start of therapy. Treatment was continued until disease progression, the patient’s decision to stop, or the appearance of unacceptable toxicity. In total, 73 mCRC patients were eligible for analysis. All samples were obtained after participants signed an informed written consent form to enter the study. The study protocol was approved by the Ethics Committee of the Reina Sofia University Hospital, Córdoba (Spain) (protocol code PI16/01271, approved on 3 February 2017, act no. 260, ref. 3404), in accordance with the fundamental principles established in the 1964 Declaration of Helsinki and subsequent amendments.

### 2.2. Clinicopathological Variables

The analyzed clinicopathological characteristics of 73 patients (Table 1) included age, sex, stage, tumor location, degree of tumor differentiation, histological tumor type, RAS-BRAF mutational status, microsatellite status, number of metastatic sites, first-line chemotherapy treatment received, duration of treatment, response assessment, and grade toxicities. Toxicity was assessed throughout the study according to the National Cancer Institute Common Toxicity Criteria.

### 2.3. Blood Collection and Plasma Separation

Plasma was obtained from 8 mL of blood collected using K2-EDTA tubes. Blood samples were centrifuged at 3.000× *g* for 10 min at 4 °C to separate plasma. Plasma samples were then aliquoted in Eppendorf tubes to avoid freeze–thaw cycles and stored at −80 °C until use. 

### 2.4. Analysis of Circulating Markers in Plasma

The measurement of the analytes in plasma was performed using the following ELISA kits: VEGF-A human ELISA kit (ref: BMS277-2, Invitrogen) and ACE human ELISA Kit (ref: ab119577, Abcam). In each case, the analysis was performed following the instructions provided by the manufacturer.

### 2.5. Statistical Analysis

IBM SPSS Statistics for Macintosh (Version 20.0, IBM Corp, Armonk, NY, USA) was used for statistical analysis. Qualitative variables were compared using the chi-squared/Fisher’s exact tests. Quantitative variables were compared using the Mann–Whitney U test. The association with survival was analyzed using a Kaplan–Meier plot and log-rank test. Since no standardized cut-off points were available for the analytes determined in plasma, the statistical analysis was performed by stratifying our population according to tertiles. Multivariate analyses for OS and PFS were performed using the Cox proportional hazards model, adjusting by age, gender, ECOG at diagnosis, RAS status, primary tumor localization, and number and localization of metastases. The significance level for all the analyses was set at *p* ≤ 0.05.

## 3. Results

### 3.1. Baseline Characteristics of Patients

From March 2017 to December 2020, 200 patients with histological confirmation of mCRC prior to the initiation of standard first-line chemotherapy treatment including bevacizumab were screened for eligibility. Among these 200 patients, 46 received chemotherapy alone (unfit patients and those not candidates for polychemotherapy); 38 received chemotherapy + anti-EGFR; 26 patients had resectable/potentially resectable disease; in 7 patients, it was not possible to determine plasma VEGFA and/or ACE; and 10 patients had a response not evaluable by RECIST criteria. As a result, 73 patients were finally included in the present study. The median follow-up time of the patients was 19 months (95% CI = 17.2–21.3).

The clinical–pathological characteristics of the patients are summarized in Table 1. The median age was 62 years, and most patients were male (60.3%). The most frequent histological subtype was adenocarcinoma (87.7%) and the majority of tumors were moderately differentiated (86.6%) and right-sided (64.4%). At diagnosis, 62 (84.9%) patients were stage IV, and 10 (12.7%) developed metastases during follow-up, with the liver being the most frequent site of metastasis (45.2%). Thirty-one (42.5%) patients underwent surgery for the primary tumor. Regarding molecular characteristics, 59 (80.8%) patients had a mutated RAS status, and 65 (89%) had stable microsatellites. The most-used chemotherapy regimen was oxaliplatin-based plus bevacizumab (86%), and 40 (54.8%) patients had a partial response. At the time of data analysis, 62 (84.9%) patients had progressed to first-line treatment and 48 (65.8%) had died.

According to the population tertiles, the cut-off points used for VEGF-A were T1: <0.1689 ng/ml, T2: 0.1690–0.4407 ng/ml, and T3: >0.4408 ng/ml; and for ACE, they were T1: <79.48 ng/ml, T2: 79.49–141.85 ng/ml, and T3: >141.86 ng/ml. No statistically significant associations were found between VEGF-A or ACE levels and the baseline characteristics of the patients (Table A1).

### 3.2. Basal VEGF-A and ACE Plasma Levels of mCRC Patients Have Prognostic Value for First-Line Treatment with Chemotherapy Plus Bevacizumab

In relation to VEGF-A plasma levels prior to the initiation of treatment with chemotherapy plus bevacizumab, and with a median follow-up of 9 months (95% CI = 8.9–11.6), the median PFS for T1 was 14.1 months (95% CI = 11–17.2), 9.1 months for T2 (95% CI = 7.2–11) and 9.7 months for T3 (95% CI = 7.6–11.9) (log-rank *p* = 0.033) (Figure 1A). Likewise, the median OS for T1 was 28.5 months (95% CI = 23.2–33.8), 22.6 months for T2 (95% CI = 18.1–27.1) and 18.3 months for T3 (95% CI = 14.6–22.1) (Log-rank *p* = 0.016) (Figure 1B). In multivariate analysis adjusted for prognostic factors (Table 2), VEGF-A remained as an independent prognostic factor for first-line treatment with chemotherapy plus bevacizumab (OS T3 vs. T1: HR 4.28, 95% CI = 1.83–9.99, *p* = 0.001; PFS T2 vs. T1: HR 2.15, 95% CI = 1.01–4.53, *p* = 0.045; PFS T3 vs. T1: HR 2.64, 95% CI = 1.21–5.75, *p* = 0.014).

No statistically significant differences were observed in the PFS analysis of patients stratified according to ACE plasma level tertiles (Figure 1C). However, those patients with ACE plasma levels in the upper tertile (>141.86 ng/ml) had a median OS of 26.37 months (95% CI = 21.5–31.17), compared to patients in T2 (23.3 months, 95% CI = 18–28.6) and those in T1 (18.1 months, 95% CI = 15.5–20.7) (log-rank *p* = 0.053) (Figure 1D). Additionally, when comparing ACE levels of T1 vs. T2 and T3, we found that overall survival was 18.1 months (95% CI = 15.5–20.7) and 25.0 months (95% CI = 21.3–28.5), respectively (log-rank *p* = 0.023). Accordingly, in the multivariate Cox regression analysis adjusting for prognostic factors (Table 3), ACE plasma levels remained as an independent factor for OS (T3 vs. T1: HR 0.44, 95% CI = 0.21–0.93, *p* = 0.032).

### 3.3. Combining VEGF-A and ACE Plasma Levels Stratifies mCRC Patients into High-Risk or Low-Risk Groups Prior to Their Treatment with Bevacizumab

Three prognostic risk groups were defined among the patients included in this study, according to ACE and VEGF-A tertiles: high risk (T1 ACE and T3 VEGF-A), intermediate risk (T2 ACE and T2 VEGF-A) and low risk (T3 ACE and T1 VEGF-A). High-risk patients had a highly significant shorter median OS compared to low-risk patients (16.6 months (95% CI = 13.2–20.0) vs. 29.8 (95% CI = 21.7–37.9), *p*-value = 0.007) (Figure 2). No significant associations were found between these prognostic risk groups and the clinical pathological variables (Table A2). However, in the multivariate Cox regression analysis adjusting for prognostic factors (Table A3), the prognostic risk groups remained as independent factors for OS.

## 4. Discussion

Bevacizumab is a humanized monoclonal antibody that is indicated as first-line treatment for mCRC in combination with chemotherapy [5], and its mechanism of action is based on its ability to bind to VEGF protein, thereby inhibiting tumor angiogenesis [24]. The predictive value of numerous biomarkers of response to anti-angiogenic drugs in mCRC, which will allow the selection of those patients with the greatest benefit and impact on survival, has been previously reviewed in the literature [25,26]. However, although functional evidence exists, none of these potential biomarkers has been shown, so far, to possess clinical value; additionally, other studies have failed to reproduce their efficacy [16,20,27]. 

In our mCRC patient cohort, we found that those patients with low basal VEGF-A plasma levels had significantly better PFS and OS rates when treated with bevacizumab, independently of other prognostic factors. Additionally, those patients with low basal ACE plasma levels displayed significantly worse OS rates after treatment with bevacizumab. Accordingly, the subgroup of patients with low VEGF-A and high ACE levels showed a significantly better OS outcome, allowing us to establish prognostic risk groups for patients who were to receive this anti-angiogenic drug in first-line treatment. 

Several studies have reported differing results on the correlation between plasma VEGF-A and outcomes in mCRC patients treated with bevacizumab [15,28]. However, the different VEGF-A assays used, as well as the fact that patients received heterogenous chemotherapy regimens, make it difficult to interpret the results of these studies [15,29,30,31]. For instance, Marisi et al. [32] found no correlation between baseline VEGF-A mRNA expression and outcomes in a randomized trial of mCRC patients receiving chemotherapy with or without bevacizumab. However, the analysis was performed in total blood, and no VEGF-A protein circulating levels were obtained, making it difficult to compare the data with other studies.

In concordance with the data reported herein, we have recently shown that circulating VEGF-A in mCRC patients is a potential biomarker to predict better outcomes in second-line chemotherapy plus the anti-angiogenic drug aflibercept [21]. Specifically, efficacy was higher in patients with lower baseline plasma VEGF-A levels, suggesting VEGF-A as a potential biomarker to predict better outcomes following aflibercept plus FOLFIRI. 

On the other hand, several studies have established an association between the RAAS and the process of angiogenesis in tumors [33]. Moreover, the arterial hypertension commonly observed during treatment with bevacizumab has been proposed as a possible biomarker of response to therapy [34,35]. ACE is a zinc metallopeptidase that catalyzes the conversion of angiotensin I to angiotensin II, playing a central role in the RAAS, which exerts important functions in the vascular system regulating blood pressure and water–electrolyte balance [36]. 

The interindividual variation in ACE levels in blood and tissues is mainly due to a common polymorphism in the ACE gene consisting of the insertion (I) or deletion (D) of a 287-bp fragment; moreover, it has been associated with risk for several diseases, including cancer [37,38]. In this regard, we have previously reported that I/D and D/D genotypes and higher (> 135 ng/mL) levels of circulating ACE were associated with better responses to bevacizumab treatment at any time point of the disease in metastatic breast cancer or CRC patients [22]. Accordingly, in the present study, we have now found that mCRC patients in the upper tertile of ACE (>141.86 ng/ml) prior to the initiation of treatment with bevacizumab had a significantly better OS. 

Finally, herein, we report a novel prognostic classification based on basal VEGF-A and ACE plasma levels in mCRC patients who will receive chemotherapy and bevacizumab as first-line therapy. We have shown that low basal levels of VEGF-A and high basal levels of ACE are associated with significantly better survival rates (16.6 months (95% CI = 13.2–20.0) vs. 29.8 (95% CI = 21.7–37.9), *p*-value = 0.007). To our knowledge, this is the first study to identify VEGF-A and ACE as potential biomarkers for selecting those mCRC patients who will most benefit from receiving chemotherapy plus bevacizumab treatment in first-line therapy. Thus, our data propose the classification of patients with ACE levels > 141.86ng/mL and VEGFA levels < 0.168 ng/mL as the most favorable prognostic group prior to the initiation of first-line treatment. This information could be useful in clinical practice, improving cost-effectiveness and therapeutic outcomes, and avoiding unnecessary toxicities.

We recognize the limitations of our study, which must be considered when interpreting these results. First, this is an observational study with a limited sample size, and without a control group that would allow us to clarify the association of these biomarkers with the response to non-anti-angiogenic chemotherapy. Additionally, we performed an analysis based on tertiles, with the aim of differentiating the groups with the greatest benefit; however, standardized cut-off points for VEGF-A and ACE must be identified in future studies. More studies are warranted to explore the evolution of these biomarkers at critical points during disease progression to clarify their true predictive and prognostic values. 

## 5. Conclusions

In summary, our study supports that VEGF-A and ACE may be useful biomarkers in the selection of mCRC patients for anti-angiogenic therapy. Additionally, our data reinforce the notion of a close association between the RAAS and the anti-angiogenic response in cancer. Further studies are needed to confirm these results and to implement the use of these biomarkers in clinical practice.

## Figures and Tables

**Figure 1 cancers-14-03054-f001:**
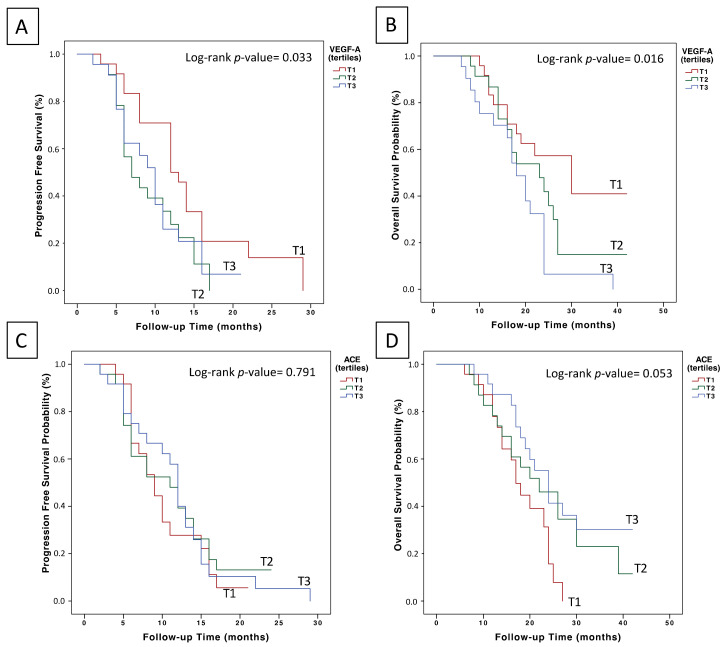
Prognostic value of basal VEGF-A and ACE plasma levels in the first-line treatment of mCRC patients with chemotherapy plus bevacizumab: (**A**) effect of basal VEGF levels on PFS outcome; (**B**) effect of basal VEGFA levels on OS outcome; (**C**) effect of basal ACE levels on PFS outcome; and (**D**) effect of basal ACE plasma levels on OS outcome.

**Figure 2 cancers-14-03054-f002:**
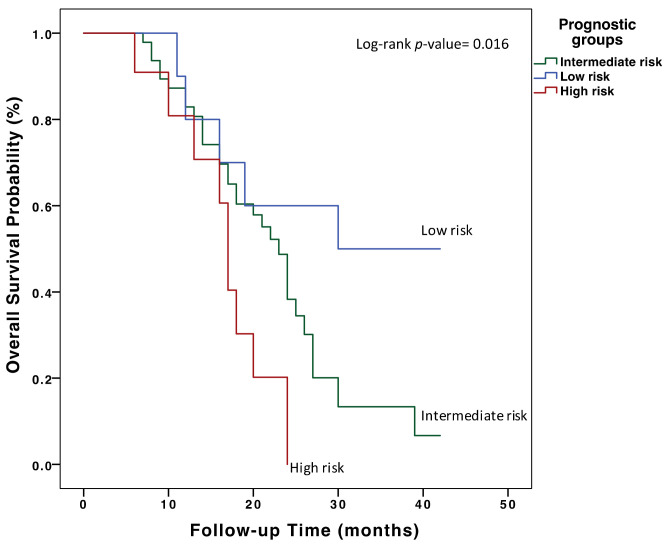
Overall survival analysis in first-line treatment of mCRC patients with chemotherapy plus bevacizumab according to basal VEGF-A and ACE plasma levels. Three prognostic risk groups were defined according to ACE and VEGF-A plasma level tertiles: high risk (T1 ACE and T3 VEGF-A), intermediate risk (T2 ACE and T2 VEGF-A) and low risk (T3 ACE and T1 VEGF-A).

**Table 1 cancers-14-03054-t001:** Clinical pathological data of patients.

Patient Characteristics		*n* (%)
Age (median, range)	62, 35–87	
Gender	Male	44 (60.3)
	Female	29 (39.7)
Localization	Right side	26 (35.6)
	Left side	47 (64.4)
Stage at diagnosis	Early stage	10 (12.7)
	Late stage	62 (84.9)
Histological subtype	Adenocarcinoma	64 (87.7)
	Mucinous/Ring cell	9 (12.4)
Histological grade	Well-differentiated	9 (12.3)
	Moderately differentiated	61 (86.6)
	Poorly differentiated	3 (4.1)
Primary tumor surgery	Yes	31 (42.5)
	No	42 (57.5)
ECOG at diagnosis	0	44 (60.3)
	1	29 (39.8)
Number of metastases	≤2	56 (76.7)
	>2	17 (23.3)
Liver metastases	Yes	33 (45.2)
	No	40 (54.8)
Lung metastases	Yes	16 (21.9)
	No	57 (78.1)
Peritoneal metastases	Yes	9 (12.3)
	No	64 (87.7)
RAS mutational status	Mutated	59 (80.8)
	Wild Type	13 (17.8)
	Unknown	1 (1.4)
BRAF mutational status	Mutated	5 (6.8)
	Wild Type	15 (20.5)
	Unknown	53 (72.6)
Microsatellite status	MSS	65 (89.0)
	MSI	2 (2.7)
	Unknown	6 (8.2)
First-line palliative chemotherapy	FOLFOX/XELOX–bevacizumab	63 (86.3)
	FOLFIRI–bevacizumab	3 (4.1)
	FOLFOXIRI–bevacizumab	4 (5.5)
	Capecitabine–bevacizumab	3 (4.1)
Response	Partial Response	40 (54.8)
	Stable disease	29 (39.7)
	Progression disease	4 (5.5)
First-line toxicity grade >2	Yes	21 (28.8)
	No	52 (71.2)
Second-line palliative chemotherapy	Yes	47 (64.4)
	No	26 (35.6)
Progression to first-line treatment	Yes	62 (84.9)
	No	11 (15.1)
Exitus	Yes	48 (65.8)
	No	25 (34.2)

**Table 2 cancers-14-03054-t002:** Multivariate analysis of OS and PFS including tertiles of VEGF-A.

Variables	OS		PFS	
	HR (95% CI)	*p*	HR (95% CI)	*p*
VEGF-A				
T1	1 (ref.)		1(ref.)	
T2	1.9	0.124	2.15	0.045
	(0.85–4.30)		(1.02–4.54)	
T3	4.28	0.001	2.64	0.014
	(1.83–10.0)		(1.21–5.65)	
Gender	0.6	0.13	1.12	0.685
	(0.31–1.16)		(0.64–1.96)	
Age	1.01	0.534	0.99	0.615
	(0.97–1.05)		(0.96–1.02)	
ECOG				
0	1 (ref.)		1(ref.)	
1	2.32	0.01	2.11	0.014
	(1.22–4.42)		(1.16–3.83)	
RAS status	0.92	0.843	0.86	0.665
	(0.40–2.11)		(0.43–1.71)	
Localization of tumor				
Right	1(ref.)		1(ref.)	
Left	0.33	0.001	0.6	0.018
	(0.17–0.65)		(0.33–1.12)	
Number and localization of metastases	0.75	0.43	0.77	0.439
	(0.36–1.55)		(0.39–1.50)	

**Table 3 cancers-14-03054-t003:** Multivariate analysis of OS and PFS including tertiles of ACE.

Variables	OS		PFS	
	HR (95% CI)	*p*	HR (95% CI)	*p*
ACE				
T1	1 (ref.)		1(ref.)	
T2	0.69	0.339	0.98	0.952
	(0.33–1.47)		(0.51–1.89)	
T3	0.44	0.032	0.95	0.879
	(0.21–0.93)		(0.50–1.80)	
Gender	0.67	0.231	1.05	0.863
	(0.35–1.29)		(0.60–1.84)	
Age	0.99	0.523	0.98	0.075
	(0.96–1.02)		(0.95–1.01)	
ECOG				
0	1 (ref.)		1 (ref.)	
1	2.14	0.021	1.97	0.025
	(1.12–4.07)		(1.09–3.58)	
RAS status	0.96	0.92	1.06	0.862
	(0.42–2.20)		(0.55–2.06)	
Localization of tumor				
Right	1 (ref.)		1 (ref.)	
Left	0.44	0.013	0.71	0.264
	(0.23–0.84)		(0.39–1.23)	
Number and localization of metastases	0.82	0.608	0.94	0.848
	(0.49–1.73)		(0.49–1.81)	

## Data Availability

The data presented in this study are available on reasonable request from the corresponding author.

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
