# Peer review of "Basal VEGF-A and ACE Plasma Levels of Metastatic Colorectal Cancer Patients Have Prognostic Value for First-Line Treatment with Chemotherapy Plus Bevacizumab"

_cancers, 2022, doi:10.3390/cancers14133054_

Round 1

Reviewer 1 Report

This manuscript describes the prognostic factors of baseline VEGF-A and ACE plasma levels in a cohort of bevacizumab-treated mcrc.  Unfortunately, no comparator group is included.  This is a shame, as the authors could have conducted this analysis in the remainder of the 200 patient cohort who did not receive bevacizumab.  The resultant combined analysis could have answered the question of whether baseline VEGF-A and ACE plasma levels are actually predictive of bevacizumab outcomes. 

There are already several published papers of non-comparative bevacizumab-treated cohorts which have demonstrated prognostic associations of various VEGF-A parameters.  

Marisi et al (2017) found no correlation between baseline VEGF-A mRNA expression and outcomes in the randomised trial of chemo +/- bevacizumab; ie VEGF-A mRNA expression was not predictive of outcome.

1.       The term “predictive” needs to be removed from the title

2.       Line 97: correct spelling “eligible”

3.       Table 1: correct spelling “late stage”

4.       Tables 2 & 3: please clarify ECOG 1 vs 2? And Localization of tumor left vs right”

5.       Results 3.2: remove “predictive” from heading

6.       Line 164: remove “predictive”

7.       Figure 1: remove “predictive from figure legend

8.       Line 221: remove “predictive” as the cited study did not have a comparator group (ie patients who did not receive aflibercept)

9.       Lines 246 and 248: remove “predictive”

Author Response

Responses to reviewer 1:

Question 1.- This manuscript describes the prognostic factors of baseline VEGF-A and ACE plasma levels in a cohort of bevacizumab-treated mcrc.  Unfortunately, no comparator group is included.  This is a shame, as the authors could have conducted this analysis in the remainder of the 200 patient cohort who did not receive bevacizumab.  The resultant combined analysis could have answered the question of whether baseline VEGF-A and ACE plasma levels are actually predictive of bevacizumab outcomes.

Answer 1.- We thank the reviewer for these comments, which allow us to clarify important points of our study. We would like to emphasize that this study was aimed to evaluate the prognostic and predictive value of response of circulating VEGFA and ACE values in a population of patients with mCRC indicated to receive bevacizumab added to standard first-line chemotherapy. Within the exclusion criteria of our study, we considered excluding those patients who were not going to receive bevacizumab for different reasons (including those who received treatment with exclusive chemotherapy and/or added to antiEGFR, more details are now provided regarding inclusion and exclusion criteria in the Material and methods section, and now we also explain in the results section why the other patients were excluded). The indication for treatment with bevacizumab is given in clinical practice according to the recommendations of the NCCN and ESMO guidelines. It is specifically in this population where we wanted to describe the effect that these markers have on PFS and OS, helping us to identify those patients who obtain the greatest benefit from this therapy. Comparing with a chemotherapy-alone group of patients, which in most cases are patients who are not candidates for intensive chemotherapy due to comorbidities or deterioration of functional status, we would not be dealing with homogeneous groups and the results would not allow to differentiate whether the effects are due to other associated unfavourable prognostic factors.

Furthermore, in our study we are not discussing the benefit of treatment with bevacizumab, but rather analyzing a marker that helps to predict which subgroup of patients will benefit most from therapy, and in this case, we consider that all patients included should receive the same treatment scheme of chemotherapy plus bevacizumab. Since we cannot analyse a population treated only with bevacizumab, and we do not have this indication in monotherapy, the predictive value of VEGFA refers to the complete scheme administered in first line with chemotherapy plus bevacizumab. Therefore, in the revised manuscript the predictive value will be always referred to chemotherapy plus bevacizumab. Accordingly, the title of the manuscript has been changed to: “Basal VEGF-A and ACE plasma levels of metastatic colorectal cancer patients have predictive and prognostic value for first-line treatment with chemotherapy plus bevacizumab”.

Question 2.- There are already several published papers of non-comparative bevacizumab-treated cohorts which have demonstrated prognostic associations of various VEGF-A parameters. Marisi et al (2017) found no correlation between baseline VEGF-A mRNA expression and outcomes in the randomised trial of chemo +/- bevacizumab; ie VEGF-A mRNA expression was not predictive of outcome.

Answer 2.- We agree with the reviewer that this is relevant information, and we now have included and discussed the work of Marisi et al (2017) in the Discussion section (lines 301-306).

Q3.- The term “predictive” needs to be removed from the title.

A3.- As discussed above (Question 1), we now specify that the predictive value is for the response to chemotherapy plus bevacizumab, and title has been changed accordingly.

Q4.- Line 97: correct spelling “eligible”

A4.- The misspelled word has been corrected.

Q5.- Table 1: correct spelling “late stage”

A5.- The misspelled word has been corrected.

Q6.- Tables 2 & 3: please clarify ECOG 1 vs 2? And Localization of tumor left vs right

A6.- We have now specified the reference groups of these variables in Table 2 & 3.

Q7.- Results 3.2: remove “predictive” from heading

A7.- As discussed above (Question 1) we now specify: “have predictive and prognostic value for first line of treatment with chemotherapy plus bevacizumab.

Q8.- Line 164: remove “predictive”

A8.- As discussed above (Question 1) we now specify: “as an independent prognostic and predictive factor for first line of treatment with chemotherapy plus bevacizumab”.

Q9.- Figure 1: remove “predictive from figure legend

A9.- As discussed above (Question 1) we now specify: “Predictive and prognostic value of basal VEGF-A and ACE plasma levels in the first line of treatment of mCRC patients with chemotherapy plus bevacizumab”.

Q10.- Line 221: remove “predictive” as the cited study did not have a comparator group (ie patients who did not receive aflibercept).

A10.- We have now re-phrased as follows: “…circulating VEGF-A in mCRC patients is a potential biomarker to predict better outcomes in second-line chemotherapy plus the antiangiogenic drug aflibercept”, to better describe the conclusions of our previous study.

Q11.- Lines 246 and 248: remove “predictive”

A11.- We have removed the term “predictive” and re-phrased as follows “To our knowledge, this is the first study to identify VEGF-A and ACE as potential biomarkers of response and survival in mCRC patients receiving chemotherapy plus bevacizumab treatment in first-line therapy.”

We would like to thank the Reviewer for his/her helpful comments, which have much improved the manuscript.

Reviewer 2 Report

Thank you for the opportunity to review this interesting manuscript. This reviewer appreciates the novel approach to assess predictive and prognostic value of basal VEGF-A and ACE plasma levels in patients with mCRC treated with bevacizumab.

The following comments and suggestions are provided for your consideration:

1.     Only 73 patients were eligible for analysis, how were the other patients excluded? Please address such exclusion might affect the generalizability of study findings.

2.     The result shown in Figure. 2 reads impressive; however, a table providing the clinical pathological data for each combination groups (e.g. group “T1 VAGF-A & T3 ACE”) would be more informative. And, multivariate analysis of OS and PFS among the combination groups (similar with table 2 and 3) is suggested.

3.     In this study, patients were grouped by tertiles. Would ROC analysis be a better approach to select the cut-off values, as employed in the POLAF trial?

4.     Suggest to move the Table 1 to the “Results”.

5.     Suggest to report the definite reference group of each variable listed in table 2 and 3.

6.  In Table 1, the sorting criteria of “Number of metastases” should be <2 and “>=2”?

Author Response

Question 1.- Only 73 patients were eligible for analysis, how were the other patients excluded? Please address such exclusion might affect the generalizability of study findings.

Answer 1.- We thank the reviewer for this comment, and now we provide more details on this important aspect. Thus, we not only describe in more detail the inclusion and exclusion criteria in the Material and methods section (lines 88-95), but also explain in the results section (3.1. Baseline characteristics of patients), why the other patients were excluded (lines 149-153).

Q2.- The result shown in Figure. 2 reads impressive; however, a table providing the clinical pathological data for each combination groups (e.g. group “T1 VAGF-A & T3 ACE”) would be more informative. And, multivariate analysis of OS and PFS among the combination groups (similar with table 2 and 3) is suggested.

A2.- Now we provide the clinical pathological data for each prognostic risk group (Table A2) and the corresponding multivariate analysis of OS and PFS (Table A3).

Q3.- In this study, patients were grouped by tertiles. Would ROC analysis be a better approach to select the cut-off values, as employed in the POLAF trial?

A3.- As described in material and methods, to date, there is no standardized cut-off values for VEGFA and ACE as circulating markers. In this study, we wanted to use a homogeneous analysis methodology, with balanced groups and given the sample size the use of tertiles is the best option. In addition, the use of a single cut off value to classify patients calculated by means of ROC curves would not allow us to evaluate the gradual biological effect that the association of VEGFA and ACE establishes on survival. With the methodology employed, our results have observed this effect when stratifying patients in tertiles of VEGFA with PFS (T1 was 14.1 months (95% CI 11-17.2), 9.1 months for T2 (95% CI 7.2-11) and 9.7 months for T3 (95% CI 7.6-11.9) (log-rank p=0.033)) and OS (T1was 28. 5 months (95% CI 23.2-33.8), 22.6 months for T2 (95% CI 18.1-27.1) and 18.3 months for T3 (95% CI 14.6-22.1) (log-rank p=0.016)), as well as CEA and OS (T3 had 26.37 months (95% CI 21.5-31.17), T2 (23.3 months, 95% CI 18-28.6) and T1 (18.1 months, 95% CI 15.5-20.7), (log-rank p=0.053). Finally, in the discussion section we have indicated the need for future studies to identify standardized cut-off values for VEGF-A and ACE.

Q4.-   Suggest to move the Table 1 to the “Results”.

A4.- We agree with the reviewer, and Table 1 has been moved to the Results section

Q5.- Suggest to report the definite reference group of each variable listed in table 2 and 3

A5.- We have now included the reference groups for variables in Table 2 & 3

Q6.- In Table 1, the sorting criteria of “Number of metastases” should be <2 and “>=2”?

A6.- Now specify the sorting criteria for the number of metastases as “£2” and “>2”

The authors would like to thank the  Reviewer for his/her helpful comments, which have much improved the manuscript.

Reviewer 3 Report

The manuscript "Basal VEGF-A and ACE plasma levels of metastatic colorectal 2 cancer patients have predictive and prognostic value for first-3 line treatment with bevacizumab" is a well written paper which investigates the potential prognostic and predictive role of circulating biomarkers for anti-angiogenic treatment. Indeed, the identification of predictive biomarkers is an unmet clinical need and would allow an improved selection of patients among those who are candidates to bevacizumab. The Authors presented their study in a very clear way and surely interesting for the readers. They also discussed their results and the limitations of their research. I believe that this manuscript is suitable for publication.

Author Response

We would like to thank the reviewer for reading our manuscript and for his/her kind comments.

Round 2

Reviewer 1 Report

Thank you for your responses, however, the authors fail to understand the difference between "prognostic" and "predictive."  In order to determine that a biomarker is predictive of response to a particular therapy, a comparator group which has not been exposed to that therapy must be used.  Without a comparator group, the biomarker might only be prognostic (ie reflect the disease process itself), without being predictive of response or better outcome with bevacizumab. 

Author Response

We again appreciate the reviewer's comments and the opportunity to improve our work. We understand the concerns about a precise distinction between "prognostic" and "predictive," so we have revised the manuscript by modifying the title and text, removing the statements about "predictive" value. We now emphasize the prognostic value of our data, and especially our identification of the association of VEGF-A and ACE as possible prognostic biomarkers in mCRC patients receiving first-line bevacizumab chemotherapy treatment. 

Reviewer 2 Report

Thank you for replying my comments and suggestions in details. I have no further questions and would like to recommend this manuscript for publication.

Author Response

We thank reviewer 2 for revising our manuscript 

Round 3

Reviewer 1 Report

Thanks for revising the manuscript